# Frenemies in the Microenvironment: Harnessing Mast Cells for Cancer Immunotherapy

**DOI:** 10.3390/pharmaceutics15061692

**Published:** 2023-06-09

**Authors:** Roberta Sulsenti, Elena Jachetti

**Affiliations:** Molecular Immunology Unit, Department of Experimental Oncology, Fondazione IRCCS Istituto Nazionale dei Tumori, 20133 Milan, Italy; roberta.sulsenti@istitutotumori.mi.it

**Keywords:** mast cells, cancer, immunotherapy, tumor microenvironment

## Abstract

Tumor development, progression, and resistance to therapies are influenced by the interactions between tumor cells and the surrounding microenvironment, comprising fibroblasts, immune cells, and extracellular matrix proteins. In this context, mast cells (MCs) have recently emerged as important players. Yet, their role is still controversial, as MCs can exert pro- or anti-tumor functions in different tumor types depending on their location within or around the tumor mass and their interaction with other components of the tumor microenvironment. In this review, we describe the main aspects of MC biology and the different contribution of MCs in promoting or inhibiting cancer growth. We then discuss possible therapeutic strategies aimed at targeting MCs for cancer immunotherapy, which include: (1) targeting c-Kit signaling; (2) stabilizing MC degranulation; (3) triggering activating/inhibiting receptors; (4) modulating MC recruitment; (5) harnessing MC mediators; (6) adoptive transferring of MCs. Such strategies should aim to either restrain or sustain MC activity according to specific contexts. Further investigation would allow us to better dissect the multifaceted roles of MCs in cancer and tailor novel approaches for an “MC-guided” personalized medicine to be used in combination with conventional anti-cancer therapies.

## 1. Introduction

Cancer development, progression, and resistance to therapy are shaped by stimuli coming from the tumor microenvironment (TME). The TME is composed of different types of cells, including fibroblasts, mesenchymal cells, immune cells, and endothelial cells, along with the presence of extracellular matrix proteins. Immune cells in the TME have the potential to attack tumor cells and therefore impede neoplastic progression. However, tumor cells can evade immune recognition and, in turn, can influence host bystander cells to support and maintain cancer growth and metastasis occurrence through a variety of different mechanisms [1]. The final outcome converges in a tumor-promoting and immune-suppressive microenvironment, as a result of constant crosstalk between cancer cells and the surrounding TME [2].

Commonly associated with allergies and autoimmune disorders, mast cells (MCs) have recently gained increasing attention in the TME, as they are endowed with both pro-angiogenic and pro-tumorigenic functions and have the ability to promote immunosuppression. However, in some instances, MCs can also actively restrain tumor growth and foster anti-tumor immunity. These opposite functions can depend on tumor type, MC peri- or intra-tumor localization, and their interaction with other immune cells.

In this review, we describe the multifaceted roles of MCs in tumor development and discuss possible therapeutic approaches that aim to target MC function for cancer immunotherapy. Still, many issues remain unresolved. The main challenge is to understand the activation state of MCs in the tumor area. The in vivo identification of different biomarkers associated with MC subtypes and activation status could better inform the development of therapeutic regimens for patients. Additionally, newly available technologies to evaluate the spatial distribution of cells within a tissue (i.e., single cell RNAseq or spatial transcriptomic) will help to deeply dissect the crosstalk between MCs and the bystander cells involved in cancer development. This comprehensive characterization would provide crucial insights into the significance of each population within specific cancer contexts.

## 2. MC Biology

MCs are cells of the innate immunity that originate from pluripotent hematopoietic cells of the bone marrow, and they mature when they reach vascularized tissues [3]. The stem cell factor (SCF)/c-Kit receptor pathway and IL-3 have a pivotal role in MC development, maturation, and proliferation [4]. SCF is also one of the main chemoattractants for MCs [5], together with CCL5, which binds to CCR1 and CCR4 [6].

MCs populate different areas of the body, such as epithelia, mucosa, gastrointestinal tracts, mucus-producing glands, and regions around nerves and blood vessels. Nevertheless, in some species, such as murine rodents, MCs are also present in pulmonary, peritoneal, and mesothelium tracts [3]. On their surface, MCs display several receptors that, once triggered by their ligands, can release a variety of different factors. These include preformed molecules (i.e., histamine, tryptases, proteases, and proteoglycans) and newly synthesized lipid mediators (i.e., leukotrienes and prostaglandins), cytokines (i.e., IL-4, TNFα, TGF-β, IL-1β), and chemokines (i.e., IL-8, CCL2, CCL4) [7]. Preformed mediators settle in large granules placed in MC cytoplasm. Every MC is endowed with about 50–200 granules that are delivered outside the cell in a few seconds upon proper stimulation. This process is called MC degranulation [8].

Murine MCs are categorized into two main classes depending on their location: mucosal-type MCs and connective tissue-type MCs. The former are characterized by the expression of MC protease (mMCP)-1 and -2, whereas the latter have mMCP-4, -5, -6, and the enzyme carboxypeptidase A (CPA). Moreover, human MCs are divided into three categories according to their serine proteases content: tryptase only (MC_T_), chymase only (MC_C_), and MCs expressing both tryptase and chymase (MC_TC_). Even though these classifications are still used, mixed phenotypes can be observed in both human and mouse MC subpopulations [9]. However, it is currently not possible to accurately determine in vivo the prevalence of each MC subpopulation in different cancer settings, and consequently, it is not possible to evaluate their specific effect on final disease outcome.

MCs are best known for their role in allergy and inflammation. The key mechanism of MC activation involves the binding of type E immunoglobulin (IgE) with its high-affinity FcεRI receptor, which triggers a huge secretory response. FcεRI is a heterotetrameric receptor formed by an IgE-binding α subunit, the membrane tetraspanin β subunit, and two disulfide-linked γ subunits that contain one immunoreceptor tyrosine-based activation motif (ITAM)—essential for the activation of the pathway. The signaling starts from the linking between IgE and FcεRI on the surface of MCs. Then, when IgE binds the cognate antigen (Ag), the aggregation of the receptor complex is induced. Consequently, Lyn is activated and phosphorylates ITAM, also activating the spleen tyrosine kinase (Syk). In turn, Lyn and Syk phosphorylate downstream proteins to induce MC activation. In addition, the FcεRI complex promotes the activation of another Src family kinase, Fyn, which phosphorylates Gab2, activating the phosphoinositide 3-kinase (PI3K) pathway [10]. This process culminates in the secretion of histamine, heparin, proteases, cytokines, and other preformed mediators present in the granules. IgE/Ag triggering also induces the de novo synthesis and release of lipid mediators, chemokines, growth factors, and cytokines [11,12].

FcεRI-dependent MC degranulation can potentially be counteracted by inhibitory receptors of the SIGLEC family, mainly SIGLEC-8 [13]. Besides IgE/FcεRI-stimulation, MC degranulation and the release of mediators can also be triggered by the MRGPRX2 receptor (MRGPRB2 in mice), almost selectively expressed by skin MCs [14]. Different compounds can activate the MRGPRX2 receptor, including somatostatin, angiopeptin, mast cell degranulating peptide (MCDP), and β-defensins [15].

MCs also have a pivotal role in innate immunity against microorganisms, as they express numerous receptors such as Toll-like receptors (TLRs; up to ten in humans and up to thirteen in mice) and nucleotide-binding oligomerization domain (NOD), which are able to identify pathogen-associated molecular patterns (PAMPs) [16]. Almost all TLRs are on the cell membrane except for TLR3, TLR7, TLR8, and TLR9, which are inside the cell. Specific TLRs can interact with others, forming heterodimers (i.e., TLR2/TLR1, TLR2/TLR6 and TLR4/TLR6), and most of them (except TLR3, TLR7, and TLR9) rely on MyD88 for their signaling pathway [17]. Notably, TLR4 stimulation can either work via MyD88 to activate NF-kB or independently to stimulate type I IFN response [18].

Other receptors present on the MC surface can lead to the release of pro-inflammatory mediators. IL-33 receptor plays a crucial role in MC biology. It belongs to the TLR/IL-1R (TIR) superfamily, which, upon stimulation, induces the secretion of several factors such as IL-1, IL-6, TNFα, CCL2, and CCL3. IL-33 signaling is also involved in the degranulation, homing, and survival of MCs [19]. Additionally, TSLPR, which triggers the release of IL-4, IL-5, IL-9, and IL-13 [20], and the receptors of the CD300 family, which can either activate (CD300c, CD300lb, CD300lh) or inhibit (CD300a, CD300f, CD300lf) cytokine production [15], represent key receptors for MC biology. MCs can also express PD-L1, a known immunocheckpoint molecule that can inhibit T cell activation [21]. Interestingly, PD-L1 blockade during the allergen sensitization phase can restrain MC degranulation [22]. As MCs are capable of releasing such a plethora of factors as a consequence of distinctive external stimuli, they are key players in several physiological and pathological mechanisms, including cancer.

## 3. MCs in Cancer

Since the first evaluation of MCs in human tumors by Paul Ehrilich, MCs have gained increased attention in cancer-related fields. Nevertheless, their role is equivocal since they can either promote or inhibit tumor development in different situations [23].

### 3.1. Pro-Tumorigenic Functions of MCs

MCs can support angiogenesis, inflammation, and homeostasis, thus supporting cancer development. In pancreatic cancer patients, accumulation of MCs within tumor lesions correlates with a dismal prognosis, whereas MCs are low or absent in normal tissues [24]. Accordingly, in the Myc-induced beta cell pancreatic cancer mouse model, tumor development and angiogenesis were reduced when MC degranulation was chemically restrained [25]. In bladder cancer, an increase in MC number is associated with high-grade lesions. Additionally, MCs can enhance bladder cancer metastasis via an ERβ/CCL2/CCR2 axis that leads to epithelial-to-mesenchymal transition (EMT) and MMP9 production by cancer cells [26]. Moreover, in thyroid cancer, MC-derived IL-8 can sustain EMT and stemness by triggering Akt phosphorylation with the consequent activation of Slug in tumor cells [27]. Additionally, the secretion of CXCL1, CXCL10, and histamine by MCs sustains thyroid cancer growth [28]. In line with these results, it has been shown that histamine also has a key role in cholangiocarcinoma. Indeed, impeding histamine secretion by MCs reduces tumor growth and neo-vessel formation [29]. Furthermore, it has been described that, in mice, either the genetic or pharmacologic inhibition of MCs induced the regression of preneoplastic polyps, thus supporting the role of MCs in colon carcinogenesis [30]. In line with these results, it was found that MCs sustain colon cancer growth, while conversely, they can help to resolve colon inflammation by fostering mucosal healing through the degradation of the alarmin IL-33 [31].

### 3.2. Anti-Tumorigenic Functions of MCs

By contrast, in other types of cancer, MCs appear to have an anti-tumorigenic role. This is the case with lung cancer [32,33] and diffuse large B-cell lymphoma [34], where the presence of MCs is a good prognostic marker.

Mechanistically, in a murine model of intestinal tumorigenesis, it was shown that MCs can induce apoptosis of adenoma cells [35]. It has also been reported that the TLR-2-mediated secretion of IL-6 is responsible for the anti-tumor function of MCs against melanoma and lung cancer, both in vitro and in vivo [36]. Moreover, in non-small cell lung cancer, MC infiltration and their production of TNFα confers improved survival among patients [37]. Indeed, anti-tumor activity mediated by TNFα derived from MCs and its cytotoxic activity was demonstrated by exploiting the WEHI-164 TNFα-sensitive cell line [38]. Furthermore, a recent pan-cancer single cell analysis of immune infiltrate in different solid tumors revealed that, in nasopharyngeal cancer, MCs have a protective role and correlate with good prognosis thanks to their production of TNFα and, in particular, their high TNFα/VEGF ratio [39]. Indeed, the same study revealed a correlation with a bad prognosis of TNFα-negative, VEGF-producing MCs in the other tumor types analyzed, including lung, colon, pancreas, and kidney [39].

### 3.3. Influence of Tumor Histotype and Localization on MC Function

Interestingly, the role of MCs can dramatically change according to different histotypes of the same tumor. This is the case with breast cancer, where MCs can alternatively promote and prevent the luminal and basal subtypes, respectively [40]. In breast cancer patients, correlation of MCs infiltration with prognosis also depends on the luminal, triple-negative, or basal-like phenotype of the tumor [41]. Similarly, in prostate cancer, we demonstrated that MCs can promote the growth of adenocarcinoma by supplying MMP9 [42] and suppressing the anti-tumor T cell response [43]; however, at the same time, they protect against aggressive neuroendocrine variants [42,44] that can emerge de novo or in resistance to hormone therapy.

Even the localization of MCs within the tumor can influence the outcome. In prostate cancer, intratumoral MCs negatively regulate angiogenesis and cancer growth, whereas peritumoral MCs support tumor development [45,46]. Indeed, intra- and peri-tumoral MCs are characterized by distinct phenotypes [47]. It has also been demonstrated that a high number of extra-tumoral MCs is associated with disease recurrence and metastasis onset after radical prostatectomy [48]. A similar association has also been observed in renal cell carcinoma patients, where peritumoral and intratumoral MCs correlate with bad and good prognosis, respectively [49,50,51]. Finally, it has been recently found that MC infiltration in tumors can be influenced by microbiome composition and that gut dysbiosis can prompt MCs towards pro-metastatic functions in a breast cancer model [52].

## 4. MC Interaction with Other Immune Cells in the TME

MCs can also regulate the function of other immune cells in the TME, thus influencing either local immunosuppression or anti-tumor immunity. For instance, in a murine hepatocarcinoma model, it has been shown that activated MCs can promote the infiltration of myeloid-derived suppressor cells (MDSCs) through the CCL2/CCR2 axis and their production of IL-17, which in turn recruits Tregs at the tumor site [53]. In the transgenic APC^Δ468^ mouse model of colon cancer, MCs can stimulate the migration of MDSCs through the production of 5-lipoxygenase, which is in itself essential to produce MC-derived leukotrienes, finally promoting intestinal polyposis [54]. We and others have also shown that MCs can increase the suppressive activity of MDSCs [55] via direct interaction through the CD40L/CD40 axis [43,56]. CD40L on MCs can also promote the expansion of IL-10-producing regulatory B cells (Breg) [57]. Moreover, by using several cancer cell lines, Huang and colleagues showed that SCF-activated MCs promote the release of adenosine that suppresses NK and effector T cells, also increasing the frequency of intratumor Treg cells [58]. Notably, it was recently demonstrated in a mouse melanoma model that MCs contribute to resistance to anti-PD-1 therapy and that their inhibition with sunitinib or imatinib efficiently synergize with anti-PD-1 toward tumor regression [59].

On the other hand, in colorectal cancer, MCs can switch the function of Tregs, which downregulate IL-10 and start to produce IL-17, thereby acquiring a pro-inflammatory phenotype [60]. Notably, MC-mediated skew of Tregs and effector T cells towards Th17 relies on the crosstalk between the OX40L/OX40 axis and the production of IL-6 [61]. However, Tregs can exploit the same OX40/OX40L axis to inhibit MC degranulation, thus making this crosstalk bidirectional [62]. Furthermore, the binding between MC-derived histamine and histamine receptor type 1 promotes Th1 polarization, whereas its signaling through histamine receptor type 2 is able to restrain both Th1 and Th2 responses [63]. Conversely, it has been found that histamine can support the immunosuppressive microenvironment by recruiting Tregs [64].

Besides the direct anti-tumor effects discussed in the previous paragraph, MC-derived TNFα is also important for T cell activation [65]. The activation and proliferation of CD8^+^ T cells are also fostered by the production of osteopontin and the expression of co-stimulatory molecules by MCs [66]. Moreover, MCs can influence the homing of effector CD8^+^ T cells toward inflammation sites via the release of leukotriene B_4_ [67]. In a murine melanoma model, it has also been observed that TLR2-activated MCs can recruit NK cells through the secretion of high doses of CCL3 [36]. In addition to CCL3, other factors secreted by MCs, such as IL-4, IL-12, and TNFα, are able to activate NK cells [68,69], and the stimulation of TLR3 or TLR9 in MCs induces the secretion of IFN- γ by NK cells [70].

The conflicting results described above suggest that MCs and their mediators can have different roles depending on the stage of cancer, their peri- or intra-tumor localization, and crosstalk with other cells of the TME. Therefore, approaches aimed at molding MC functions could represent effective strategies for cancer immunotherapy.

## 5. Therapies Aimed at Targeting MCs in Cancer

The targeting of MC functions has been widely exploited for therapeutic purposes in allergic reactions and mastocytosis, as comprehensively reviewed elsewhere [71,72]. Given the focus of this review, here we discuss strategies that only specifically address the roles of MCs in cancer.

As MCs can exert tumor-promoting or suppressive activities depending on tumor type, localization, and signals received form the surrounding microenvironment, therapeutic strategies could either be directed to abrogate or prompt MC functions in the appropriate settings. Several approaches have been proposed (Figure 1): (1) targeting c-Kit signaling; (2) stabilizing MC degranulation; (3) triggering activating/inhibiting receptors; (4) modulating MC recruitment; (5) harnessing MC mediators; (6) adoptive transferring of MCs.

### 5.1. Targeting c-Kit Signaling

As c-Kit is crucial for MC development, survival, and activation, tyrosine kinase inhibitors, such as imatinib, nilotinib, or dasatinib, are efficiently employed to target MCs in mastocytosis, arthritis, or allergic responses [73,74,75,76]. However, so far, these drugs have displayed limited application in the context of restraining MC tumor-promoting functions [71]. We reported that imatinib can slightly control MC number and degranulation, consequently hampering prostate adenocarcinoma [44]. However, we found that imatinib treatment in the TRAMP prostate cancer model resulted in the outgrowth of the neuroendocrine prostate tumor variant as a drawback. This warns us about the mere targeting of MCs in prostate cancer, unless specific for adenocarcinoma-promoting functions of MCs or unless coupled with strategies designed to target the neuroendocrine histotype [44]. Another study employing the 4T1 breast cancer model showed that imatinib and the MC stabilizer cromolyn similarly induced peri-tumoral blood clotting and promoted tumor growth [77]. On the contrary, in a preclinical model of melanoma, it has recently been demonstrated that MC targeting with imatinib or sunitinib increased the therapeutic efficacy of anti-PD-1 [59].

Nevertheless, it is worth highlighting that imatinib, nilotinib, or dasatinib are not specific for c-Kit, as they also target other kinase receptors such as PDGFR, Src, and Abl kinase and thus could have off-target effects. To overcome these limitations, a monoclonal antibody targeting c-Kit, barzolvolimab, has been developed but tested so far only in the context of c-Kit-positive gastrointestinal tumors [71] or in chronic urticaria [78].

### 5.2. Stabilizing MC Degranulation

Agents that can restrain MC degranulation, such as cromolyn sodium salt or ketotifen, have been widely utilized in the treatment of allergic responses [79]; however, so far, their administration for cancer therapy has been confined to preclinical models [42,80,81]. A huge limitation of the use of “MC stabilizers” in cancer is the fact that they are not specific for MCs but can affect the release of mediators by other types of cells. In most cases, this could be an advantage if the tumor cell itself represents the target [82,83]. Nevertheless, as immune cells are also able to secrete a plethora of different mediators, it would be a disadvantage when the inhibition of T cell function occurs as a result [84].

A more tailored strategy relies on drugs targeting the intracellular components of the IgE/FcεRI signaling pathway, including Syk, PI3K p110δ isoform, and Bruton’s tyrosine kinase (BTK). The use of inhibitors of Syk and PI3K p110δ with the purpose to target MCs has so far only been investigated in the context of allergic disease [85]. Conversely, the BTK-specific drug ibrutinib was successfully tested in a model of pancreatic cancer, where it reduced tumor growth by reducing MC-dependent fibrosis [71]. However, these signal transducers are also not exclusively expressed by MCs; thus, in vivo effects could be altered by off-target activity and/or toxicity.

Recently, we discovered a new way to prevent MC degranulation by utilizing the antiepileptic drug levetiracetam [86], which is administered to avoid and mitigate seizures in patients [87]. Levetiracetam targets the synaptic vesicle protein 2A (SV2A) [88]—mainly present in neural and endocrine cells and involved in cell exocytosis [89]—and we found that this protein is also expressed by prostate cancer-infiltrating MCs [86]. We demonstrated that levetiracetam can inhibit MC degranulation and, in particular, the release of MMP9, thus restraining the pro-adenocarcinoma activity of MCs in TRAMP mice [86]. Furthermore, we also found the expression of SV2A in neuroendocrine prostate cancer and proved that, in this preclinical model, levetiracetam was also able to prevent neuroendocrine differentiation of adenocarcinoma after hormone therapy [86].

### 5.3. Triggering Other Activating/Inhibiting Receptors

Besides c-Kit and FcεRI, MCs have a plethora of different receptors that can regulate their functions in the TME; thus, these receptors could be possible targets of MC-specific anti-cancer therapies. Nevertheless, for most of them, including TSLPR, MRGPRX2, CD300, and SIGLECs, therapeutic targeting has been so far addressed only in the context of allergic diseases and extensively reviewed elsewhere [71,72,90,91].

Stimulation of TLRs in MCs can lead to specific cytokine secretion and, consequently, to the recruitment and activation of immune cells, eventually inhibiting tumor growth. Indeed, TLR agonists are currently being evaluated in cancer therapy to activate the immune response [92,93,94], and a few reports have also demonstrated the efficacy of polarizing MCs to orchestrate anti-tumor immunity. In the B16.F10 mouse melanoma model, the TLR2 agonist Pam_3_CSK_4_ induced MCs to release cytokines such as IL-6 and CCL3, which mediated a direct antiproliferative effect on tumor cells and the recruitment of NK and T cells, respectively [36]. Another study in melanoma models and patients showed a correlation between an LPS-related signature and responsiveness to anti-CTLA4 immunocheckpoint therapy. Further investigation showed that LPS triggered MCs to release CXCL10 with consequent T cell recruitment, also proving the superior anti-tumor activity of adding intratumoral LPS administration to anti-CTLA4 treatment in the preclinical setting [95]. TLR agonists given as adjuvants in the context of anti-tumor vaccinations yielded greater anti-tumor immune responses. This effect was found to be directly mediated by MCs in a mouse colon carcinoma model treated with a vaccine formulated with a recombinant CEA IgV N domain and the TLR3 ligand poly I:C [96]. Another work in the B16.F10 model also demonstrated that TLR7/9 stimulation with imiquimod can trigger CCL2 release by MCs to efficiently recruit plasmacytoid DCs at the tumor site, which, in turn, can directly kill tumor cells via TRAIL and granzyme B [97]. On the same line, it was also shown that MCs (the MC/9 cell line) pre-treated with imiquimod could increase the expression of DC costimulatory and activation molecules, thus augmenting the efficacy of a DC-based vaccine when tested in vivo in the B16-OVA melanoma model [98].

Even if TLR stimulation can efficiently reprogram MCs to orchestrate anti-tumor immunity, it should be kept in mind that this strategy could also have the opposite effect. Indeed, it was demonstrated that co-stimulation with SCF and TLR4 ligands induced the expression of VEGF, PDGF, and IL-10 in MCs, thus enhancing their tumor-promoting function in vitro and in vivo [99]. Therefore, further investigation is required to best dissect the effect of TLR triggering or inhibition to mold MCs toward anti-tumor functions.

Other possible strategies could focus on other receptors shown to be essential for the interaction of MCs with immune suppressive cells, including, for example, CD40L and OX40L, which we and others have identified as essential for the crosstalk between MCs and MDSCs [43,56], Breg [57], and Treg cells [61,62]. MCs can also directly suppress CD8^+^ T cell activation via PD-L1 on their surface [21]. Thus, MCs could be another target of immunocheckpoint blockade therapy in those tumors in which they are enriched. Notably, the inhibition of MC-associated PD-L1 resulted in increased T cell activation and efficient blunting of tumor growth in a gastric carcinoma model [100].

### 5.4. Modulating MC Recruitment

According to the pro- or anti-tumor functions exerted by MCs in different contexts, one conceivable therapeutic strategy would rely on inhibiting or incrementing their recruitment, respectively, by actioning chemotactic pathways. Besides regulating MC maturation, proliferation, and degranulation, both the SCF/c-Kit and the FcεRI signaling can also mediate MC migration [101]. Additionally, different types of tumor cells are known to produce SCF and actively recruit MCs [58,102]. Therefore, [103] inhibitors of c-Kit, BTK, Syk, and PI3K could also restrain MC trafficking in the tumor [104].

Furthermore, many other different molecules produced by tumor cells or by cells of the TME can induce MC chemotaxis, including CCL2 [105], CCL5 [106], CCL11 [106], CCL15 [102], CXCL12 [107], VEGF [80], FGF2 [108], osteopontin [109], and lipid mediators [103]. The blocking of these chemoattractants could represent a therapeutic strategy to impede MC recruitment and, consequently, their support to the tumor. In a murine model, it has been shown that the hindering of MC infiltration by treatment with the CXCL12 inhibitor AMD3100 restrains pancreatic cancer growth [110]. Similarly, in the TRAMP-C2 prostate cancer transplantable model, the blocking of FGF with the NSC12 molecule inhibited tumor growth and vascularization, and the effect was correlated to a reduced number of infiltrating MCs [108]. Additionally, in a different set of prostate cancer preclinical models, the pharmacologic inhibition of protein kinase D blunted the production of SCF, CCL5, and CCL11 and led to decreased MC migration and reduced tumor growth in vivo [106].

On the other hand, in those settings where MCs exert anti-tumor activity, it would be desirable to increase their number via the local delivery of specific chemoattractants or by stimulating the TME to produce such molecules. These approaches require further investigation.

### 5.5. Harnessing MC Mediators

As described in the previous sections, MCs can release several molecules with tumor-promoting or tumor-inhibiting functions. Thus, the last way to shape MC functions in cancer could rely on the modulation of these mediators.

Similar to many other cells in the TME, MCs secrete VEGF to promote angiogenesis. Several approved drugs against VEGF, including both monoclonal antibodies and small molecule inhibitors, are clinically available for cancer patients, as extensively reviewed elsewhere [111]. Moreover, tryptase is endowed with well-known pro-angiogenic functions [112]. Indeed, it has been described that the pharmacologic inhibition of tryptase with nafamostat or APC366 restrained tumor growth in preclinical models of pancreatic [113] and breast cancer [114], respectively. MCs also represent a major source of MMPs, which can degrade the extracellular matrix, favoring tumor growth. Nevertheless, the promising results obtained in preclinical models were not confirmed in clinical trials, where MMP inhibition was also accompanied by unexpected severe side effects [115].

Histamine can either promote or suppress anti-tumor immunity depending on tumor type and the surrounding TME [64,65,116]. Numerous studies have investigated the administration of histamine or of antagonists of its receptors in combination with immunotherapeutic strategies, showing promising results in different cancer types, including leukemia, melanoma, kidney, colorectal, and prostate cancer (reviewed in ref. [116]).

On the other hand, IL-6 and TNFα are the main molecular mediators of the anti-tumor functions of MCs [36,37,38,39], with the latter able to directly activate either apoptosis or necroptosis pathways in tumor cells [117,118]. Nonetheless, the systemic delivery of pro-inflammatory cytokines causes high toxicity [119,120,121]; thus, it is not suitable in a clinical setting. To overcome these limitations, several strategies to reduce systemic toxicity and implement the anti-tumor efficacy of TNFα have been investigated, including local administration [122] and specific delivery via nanoparticles [123,124] or targeting antibodies/peptides [125,126]. In this regard, targeting of TNFα to tumor neo-vasculature by conjugation with a CNGRC peptide that binds the CD13 aminopeptidase N [127] revealed a strong anti-tumor efficacy in several preclinical models [128], and it is being evaluated in clinical trials in melanoma and other solid tumors [129,130]. A novel frontier for localized TNFα delivery could also potentially be implemented via the direct injection of TNFα-producing MCs into the tumor. Indeed, the adoptive transfer of MCs could represent a new type of cellular immunotherapy, as described below.

### 5.6. Adoptive Transferring of MCs

Adoptive cell transfer exploiting autologous T cells or CAR-T cells is now mainstream for hematologic malignancies [131] and in widespread use for solid tumors [132]. Nevertheless, many other non-T immune cells can exert anti-tumor functions and are currently being investigated for cellular immunotherapy, including NK cells [133] and macrophages [134]. In this view, the in vivo adoptive transfer with MCs (also named reconstitution) has been exploited to demonstrate MC-specific tumor-promoting or tumor-inhibiting functions by means of reconstituting MC-deficient mice with bone marrow-derived MCs either proficient or deficient for selected molecules [55,57,135,136,137,138]. Nonetheless, it is conceivable to exploit the anti-tumor properties of MCs for cellular therapy against cancer. Such an approach should take into account the need to reprogram MCs in order to release anti-tumor mediators (i.e., TNFα) only when in contact with tumor cells to avoid the systemic delivery of unwanted molecules, allergic reactions, or other side effects. A first attempt in this direction was made by Fereyoduni et al., who exploited MCs pre-sensitized with HER2/neu-specific IgE to efficiently kill HER2/neu-expressing tumor cells both in vitro and in vivo in a breast cancer model. Indeed, they showed that the encounter with the cognate antigen on tumor cells unleashed the release of TNFα by IgE-presensitized MCs, eventually inducing apoptosis in tumor cells [139].

More investigations are required to best exploit the potential of MCs for adoptive cell therapies in cancer and also to evaluate whether it is possible to engineer MCs before in vivo transfer, with the aim of increasing and/or restraining the production of anti- or pro-tumor mediators, respectively [140]. Another strategy worthy of study would rely on exploiting defined activating receptors (i.e., TLRs) on the surface of MCs instead of IgE/FcεRI activation in order to trigger the release of desired mediators when cognate ligands are expressed by tumor cells or the surrounding TME.

## 6. Clinical Trials Aimed at Targeting MCs in Cancer Patients

Despite the intense preclinical investigation, the number of clinical trials evaluating MCs in cancer is scarce. A search on clinicaltrials.gov using #cancer and #mast cells as keywords retrieved 130 results, which were reduced to 4 after filtering to exclude trials in patients with mastocytosis and mast cell leukemias. In particular, the NCT05076682 phase II trial will test the combination of cromolyn and anti-PD1 antibodies in order to see whether the inhibition of MCs can synergize with immunocheckpoint blockade in triple-negative breast cancer patients. This trial is still in the recruiting phase and is the only one among the few identified that aims to actively target MCs with specific therapy. Two other trials in lung cancer and basal cell carcinoma (NCT02161523 and NCT02576769) list MCs among the outcomes of the study. The first will monitor MC activation/phenotype after co-culture with patient-derived fibroblasts, whereas the second is an observational study that will analyze the number of MCs in the TME. Finally, we include in the list a phase I trial that tested a therapeutic antibody directed against the folate receptor in different solid tumors, including tumors in the kidney, endometrium, lung, breast, bladder, colon, and pancreas. As the antibody used for therapy regarding the IgE isotype, it is conceivable to hypothesize the induction of some degree of MC activation. Yet, MCs are not listed among the cells that will be evaluated as readout after therapy.

## 7. Limitations

This review aimed to describe the different roles of MCs in cancer and the possible ways to exploit them for cancer immunotherapy. For this reason, we did not cover additional aspects of MC biology that are already known to be relevant in other physiological or pathological settings. For example, we did not mention that, similar to neutrophils, MCs are able to protrude extracellular traps during pathogenic infections or that MCs can be involved in fibrosis and heart failure. Further investigation is required to determine whether such MC functions are also conserved in the TME.

Furthermore, whereas different subtypes of human and murine MCs can be defined by their protease expression, their actual identification within tumor tissues is biased by some technical limitations. Thus, the relevance of MC subsets in determining pro- or anti- tumor effects is underexplored. It would be highly valuable to exploit new methodologies that can discriminate these subpopulations in the TME by evaluating their chymase/tryptase levels or new biomarkers. This would provide crucial insights into the significance of each population within specific cancer contexts.

## 8. Conclusions and Future Directions

MCs are crucial players in the TME; thus, they are potential targets for anti-tumor immunotherapy. However, extensive investigation would be required before being able to efficiently apply MC-directed therapies for the benefit of cancer patients. Indeed, given the multifaceted roles of MCs in different contexts, approaches should aim to either inhibit or foster their activity according to the specific context, contributing to the notion of “MC-guided” personalized medicine. Moreover, tailored strategies that only target specific MC functions should be preferred to avoid off-target effects or paradoxical effects, as shown for prostate cancer, where MCs show split mechanisms to either promote or prevent different tumor histotypes. The harnessing of MC functions should also be tested in combination with other (immuno-) therapeutic approaches to evaluate possible synergisms against tumor growth. Finally, evidence collected in murine models should be extensively validated in human settings to ultimately prove the clinical relevance and effective benefit of MC-based immunotherapy in cancer patients.

## Figures and Tables

**Figure 1 pharmaceutics-15-01692-f001:**
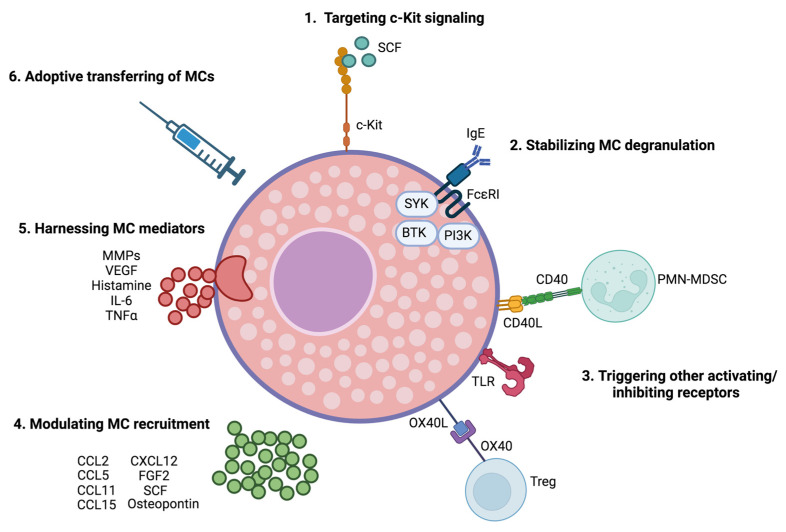
Strategies for MC-based immunotherapy in cancer. As described in the text, approaches to address MCs functions in cancer can rely on: (1) targeting c-Kit signaling; (2) stabilizing MC degranulation; (3) triggering activating/inhibiting receptors; (4) modulating MC recruitment; (5) harnessing MC mediators; (6) adoptive transferring of MCs. As MCs can exert tumor-promoting or suppressive activities depending on tumor type, their localization, and signals received from the surrounding microenvironment, therapeutic strategies could be either directed to abrogate or prompt MC functions according to the specific context. This picture was created with BioRender.com.

## Data Availability

Data sharing is not applicable to this article as no new data were created or analyzed in this study.

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
