# Peer review of "Frenemies in the Microenvironment: Harnessing Mast Cells for Cancer Immunotherapy"

_pharmaceutics, 2023, doi:10.3390/pharmaceutics15061692_

Round 1
Reviewer 1 Report
Sulsenti & Jachetti reviewed the anti- and pro-tumorigenic properties of mast cells in cancer, and its strategy in cancer immunotherapy. This review is comprehensively and structurally written. I indeed enjoy reading your manuscript. Several comments are required the authors' attention before it can be published.
1) There are several similar reviews on the mast cells and cancer immunotherapy recently (DOI: 10.3390/cells10061270. What are the strengths of this review that make it deserve to be published?
2) What are the limitations of this review?
3) Any reason for not going for a systematic review?
4) Suggest having a subheading like "pro-tumorigenic" and "anti-tumorigenic" for section 2: MC biology. May considering this for section 3: MCs in cancer.
5) What are the technical difficulty or limitations (or research gaps) in MC cancer immunotherapy research? Perhaps the authors can make this as 1 paragraph.
6) Understand that the relevance of clinical trials of mast cells (or as adjuvant) can be scarce. Possible for the authors to describe and perhaps list down all the reported or ongoing clinical trials. Kindly update the preliminary outcomes on the MC efficacy and effectiveness (or safety).
Thank you
Author Response
Sulsenti & Jachetti reviewed the anti- and pro-tumorigenic properties of mast cells in cancer, and its strategy in cancer immunotherapy. This review is comprehensively and structurally written. I indeed enjoy reading your manuscript. Several comments are required the authors' attention before it can be published.
We thank the Reviewer for the constructive suggestions regarding our manuscript.
1) There are several similar reviews on the mast cells and cancer immunotherapy recently (DOI: 10.3390/cells10061270. What are the strengths of this review that make it deserve to be published?
We are aware of the existence of several reviews that well describe the role of mast cells in cancer and the novel approaches to target them for anti-tumor response. Yet, most of these reviews are focused on ways of blocking mast cell functions or the effects induced by their mediators.
Our work explores the different therapeutic options not only to defeat pro-tumor functions of mast cells but also to enhance their anti-tumoral effect. Indeed, we propose the injection of mast cells for therapeutic purposes. Also, we suggest that the reprogramming of mast cells before the adoptive transfer could be another strategical option to overcome systemic delivery of unnecessary molecules, allergic reactions or side effects. Our point of view is to improve the vision of the powerful side of mast cell as anti-tumor cells able, in the appropriate setting, to secrete soluble factors that block cancer development, as for example IL-6 and TNFa.
2) What are the limitations of this review?
This review has the purpose to describe the different roles of MCs in cancer, and the possible ways to exploit them for cancer immunotherapy. For this reason, we did not cover additional aspects of MC biology that so far are known to be relevant in other physiological or pathological settings. For example, we did not mention that, like neutrophils, MCs are able to protrude extracellular traps during pathogenic infections, or that MCs can be involved in fibrosis and heart failure. Future investigation is required to unveil whether such MC functions are conserved also in the tumor microenvironment.
Furthermore, whereas different subtypes of human and murine MCs can be defined by their protease content, their actual identification within tumor tissues is biased by some technical limitations. Thus, the relevance of MC subsets in determining pro- or anti- tumor effects is underexplored. It would be highly valuable to exploit new methodologies that can differentiate between these sub-populations within the tumor, evaluating their chymase/tryptase levels, or new biomarkers. This would provide crucial insights into the significance of each population within specific cancer contexts.
We added these considerations in the new paragraph 7 of the manuscript.
3) Any reason for not going for a systematic review?
We think that a systematic review would have been overwhelming and challenging for a general audience. Instead, our goal with this manuscript was to offer a perspective on the various strategies used to target mast cell functions in cancer therapy. Our intention was to engage and captivate readers who may not be well-versed in the subject matter. By providing concise yet adequate information, our aim was to stimulate readers' interest and direct them towards more specialized literature for further exploration.
4) Suggest having a subheading like "pro-tumorigenic" and "anti-tumorigenic" for section 2: MC biology. May considering this for section 3: MCs in cancer.
Section 2 describes general aspects of MCs biology not related to cancer. We have introduced the suggested sub-headings in section 3, which specifically addresses MCs functions in cancer.
5) What are the technical difficulty or limitations (or research gaps) in MC cancer immunotherapy research? Perhaps the authors can make this as 1 paragraph.
We thank the Reviewer for this suggestion. The main technical limitation in studying MCs in cancer is to understand the activation state of MCs into the tumor area. The in vivo identification of different biomarkers associated with MC subtypes and activation status could better inform the development of therapeutic plans for patients. Also, new available technologies aimed to evaluate spatial distribution of cells within a tissue (i.e., single cell RNAseq or spatial transcriptomic) will help to deeply dissect the cross-talk between MCs and bystander cells involved in cancer development. This comprehensive characterization would provide crucial insights into the significance of each population within specific cancer contexts.
We added this part in paragraph 1 of the manuscript.
6) Understand that the relevance of clinical trials of mast cells (or as adjuvant) can be scarce. Possible for the authors to describe and perhaps list down all the reported or ongoing clinical trials. Kindly update the preliminary outcomes on the MC efficacy and effectiveness (or safety).
The reviewer is right in thinking that the number of clinical trials evaluating mast cells in cancer is very limited. A search on clinicaltrials.gov using #cancer and #mast cells as keywords retrieved 130 results. However, filtering results excluding trials involving mastocytosis and mast cell leukemias, we identified only 4 trials in solid tumors that intend to target or analyze MCs. In particular, the NCT05076682 phase II trial will test the combination of cromolyn and anti-PD1 antibodies in order to see whether inhibiting MCs can synergize with immunocheckpoint blockade in triple negative breast cancer patients. This trial is still in the recruiting phase, and is the only one among the few identified that aims to actively target MCs with specific therapy. Two other trials in lung cancer and basal cell carcinoma (NCT02161523 and NCT02576769) indicate MCs among the outcomes of the study. The first will monitor MCs activation/phenotype after co-culture with patient-derived fibroblasts, whereas the second is an observational study that will analyze the number of MCs in the TME. Finally, we include in the list a phase I trial that tested a therapeutic antibody directed against the folate receptor in different solid tumors including kidney, endometrium, lung, breast, bladder, colon and pancreas. Being the antibody used for therapy of the IgE isotype, it would be conceivable to hypothesize the induction of some degree of MCs activation. Yet, MCs are not listed among the cells that will be evaluated as readout after therapy.
We added this part in the new paragraph 6 of the manuscript.
Reviewer 2 Report
After careful review, I might write that the current manuscript is an interesting review discussing mast cells as an important component of tumor microenvironment. It is a well organized and clearly presented manuscript. It discussed also possible immuno-therapeutic strategies involving mast cells. The current manuscript might be published after minor editing to enhance language.
The language is fine in general. Only minor editing might be addressed to to enhance language.
Author Response
After careful review, I might write that the current manuscript is an interesting review discussing mast cells as an important component of tumor microenvironment. It is a well organized and clearly presented manuscript. It discussed also possible immuno-therapeutic strategies involving mast cells. The current manuscript might be published after minor editing to enhance language.
We want to thank the reviewer for the positive consideration of our manuscript.
The language has been checked by a native English-speaking colleague.
Reviewer 3 Report
Dear authors,
Thanks for your contribution on this field. This is an interesting state of art on mast cells (MC) and their role in cancer. The manuscript is still well written and organized.
All the aspects are cover.
Indeed, I just have a question. In the introduction part (line 62-69), different subtypes of MC are described. Since MC can have different effects on tumor growth (exacerbating or inhibiting), is there a link with those different phenotypes and effects?
Also, to complete the manuscript, it will be of interest to add information on clinical trials if there are some.
All the best,
Author Response
Dear authors,
Thanks for your contribution on this field. This is an interesting state of art on mast cells (MC) and their role in cancer. The manuscript is still well written and organized.
All the aspects are cover.
We thank the reviewer for the kind evaluation and suggestions.
Indeed, I just have a question. In the introduction part (line 62-69), different subtypes of MC are described. Since MC can have different effects on tumor growth (exacerbating or inhibiting), is there a link with those different phenotypes and effects?
Whereas different subtypes of human and murine MCs can be defined by their protease content, it is currently not possible to accurately determine in vivo the prevalence of one population over the other in different cancer settings and consequently it is not possible to evaluate their specific effect on final disease outcome. It would be highly valuable to exploit new methodologies that can differentiate between these sub-populations. This would provide crucial insights into the significance of each population within specific cancer contexts.
These considerations have been included in the manuscript (paragraph 1 and paragraph 8)
Also, to complete the manuscript, it will be of interest to add information on clinical trials if there are some.
In addressing the request of this Reviewer and of Reviewer #1, we included in the manuscript a new paragraph listing the few clinical studies that target/evaluate MCs in cancer. Yet, the number of clinical trials evaluating mast cells in cancer is very limited. A search on clinicaltrials.gov using #cancer and #mast cells as keywords retrieved 130 results. However, filtering results excluding trials involving mastocytosis and mast cell leukemias, we identified only 4 trials in solid tumors that intend to target or analyze MCs. In particular, the NCT05076682 phase II trial will test the combination of cromolyn and anti-PD1 antibodies in order to see whether inhibiting MCs can synergize with immunocheckpoint blockade in triple negative breast cancer patients. This trial is still in the recruiting phase, and is the only one among the few identified that aims to actively target MCs with specific therapy. Two other trials in lung cancer and basal cell carcinoma (NCT02161523 and NCT02576769) indicate MCs among the outcomes of the study. The first will monitor MCs activation/phenotype after co-culture with patient-derived fibroblasts, whereas the second is an observational study that will analyze the number of MCs in the TME. Finally, we include in the list a phase I trial that tested a therapeutic antibody directed against the folate receptor in different solid tumors including kidney, endometrium, lung, breast, bladder, colon and pancreas. Being the antibody used for therapy of the IgE isotype, it would be conceivable to hypothesize the induction of some degree of MCs activation. Yet, MCs are not listed among the cells that will be evaluated as readout after therapy.
We added this part in the new paragraph 6 of the manuscript.
Reviewer 4 Report
I found the manuscript titled "Frenemies in the microenvironment: harnessing mast cells for cancer immunotherapy" by Sulsenti and Jachetti to be highly intriguing. The Review article is well-written, and the topic is both current and vital in the field of cancer research. While mast cells have long been recognized as crucial players in allergic diseases, their involvement in the development and progression of cancers with unrelated histologic origins has emerged in recent years. The Authors elucidate the role of these immune cells within the complex microenvironment. Furthermore, the article delves into the discussion of mast cells' potential in cancer immunotherapy. The abundance of references provided in the manuscript allows readers to gain a comprehensive understanding of this particular research domain.
The Article is well-written and quality of English is good.
Author Response
I found the manuscript titled "Frenemies in the microenvironment: harnessing mast cells for cancer immunotherapy" by Sulsenti and Jachetti to be highly intriguing. The Review article is well-written, and the topic is both current and vital in the field of cancer research. While mast cells have long been recognized as crucial players in allergic diseases, their involvement in the development and progression of cancers with unrelated histologic origins has emerged in recent years. The Authors elucidate the role of these immune cells within the complex microenvironment. Furthermore, the article delves into the discussion of mast cells' potential in cancer immunotherapy. The abundance of references provided in the manuscript allows readers to gain a comprehensive understanding of this particular research domain.
We thank the reviewer for the positive evaluation of our manuscript.
As suggested, the language has been checked by a native English-speaking colleague.